# Effect of Eggshell Powder on the Hydration of Cement Paste

**DOI:** 10.3390/ma12152483

**Published:** 2019-08-05

**Authors:** Natnael Shiferaw, Lulit Habte, Thriveni Thenepalli, Ji Whan Ahn

**Affiliations:** 1Korea Research Institute on Climate Change, 11, Subyeongongwon-gil, Chuncheon-si 24239, Korea; 2Department of Resources Recycling, University of Science & Technology, 217 Gajeong-ro, Gajeong-dong, Yuseong-gu, Daejeon 34113, Korea; 3Center for Carbon Mineralization, Mineral Resources Division, Korea Institute of Geosciences and Mineral (KIGAM), 124 Gwahagno, Gajeong-dong, Yuseong-gu, Daejeon 34132, Korea

**Keywords:** utilization, eggshell powder, cement, hydration, monocarboaluminate

## Abstract

Eggshells are one of the solid wastes in the world and are considered hazardous according to European Commission regulations. The utilization of solid wastes, like eggshells, will help create a sustainable environment by minimizing the solid wastes that are disposed into the environment. The utilization of eggshell powder in cement also helps to reduce the carbon dioxide emissions from cement factories by reducing clinker production. In this study, the effect of eggshell powder on the hydration of cement products was investigated using X-ray diffraction (XRD), thermogravimetric analysis (TGA), Fourier-transform infrared spectroscopy (FT-IR), and scanning electron microscopy (SEM). Pastes were made with 10% and 20% eggshell powder and examined for 1, 14, and 28 days of hydration. The addition of eggshell powder transformed ettringite to monosulfoaluminate and to monocarboaluminate. In 20% eggshell powder, the formation of monocarboaluminate was detected in the early stages and accelerated the hydration reaction. The CaCO_3_ from the eggshells reacted with the C_3_A and changed the hydration products of the pastes. The addition of eggshell powder provided nucleation sites in the hydration products and accelerated cement hydration.

## 1. Introduction

Cement is the second most important material consumed daily in human life next to water. It is mandatory for socioeconomic growth and the development of countries. The manufacturing of cement is immensely energy intensive. It is also a major cause of environmental problems and accounts for around 5% of global carbon dioxide emissions [1]. The control of wastes is also currently one of the challenges of countries. The improper management of solid waste leads to public health and environmental problems [2]. Using industrial byproducts and wastes as a partial replacement of cement will reduce the emission of carbon dioxide from the production of clinker, reduce energy consumption by producing less clinker, and help the management of wastes. Fly ash is among the byproducts of coal combustion. The use of supplementary cementitious materials (SCMs) like fly ash will also help in ecological and technological aspects [3]. Grzegorz et al. [4] described how fly ash has an impact on ecology by reducing problematic landfill sites, CO_2_ emissions into the atmosphere, and the energy produced during cement production. One of the wastes is eggshells from chicken factories, egg farmers, households, and egg breaking plants. According to studies, eggshell waste generation is 150,000, 190,000, and 11,000 tons annually in the United States, India, and the United Kingdom, respectively. In the United States, companies dispose of waste eggshells in landfills at a cost of approximately $100,000 annually [5]. In some countries, they are unwilling to dispose of eggshells in landfills because the membrane that is attached to the eggshells attracts vermin and causes the spread of diseases. Figure 1 shows the waste eggshells that have been sent to landfills in Edmonton, Canada in 2017 [6].

Eggshells are one of the sources of calcium carbonate (CaCO_3_), which can replace limestone. It has been concluded that eggshells contain 96–97% of CaCO_3_, in which 3–4% is organic matter [7]. Eggshells have a pure and more stable form of CaCO_3_ called calcite, whereas limestone may contain impurities such as sand, clay, and other minerals. Limestone in the form of powder can be used in many applications such as a filler in cement [8], concrete [9], bricks [10], and for the removal of heavy metals from water [11]. In Europe, Portland limestone cement is widely used. According to European standard EN 197-1, calcium carbonate can be used up to 35% [12]. Bonavetti et al. [8] studied the effect of limestone as a filler material in cement and showed an increase in the degree of hydration with a very low w/c ratio and high limestone filler content, and the highest volume of hydration product occurred for high w/c ratio pastes. However, the compressive strength at 28 days was reduced. Pliya et al. [9] studied the effect of eggshells, which are used as a replacement in Portland cement, and showed a decrease in the compression and flexural strengths. Mtallib et al. [13] investigated the effect of eggshell ash on the setting time of cement and concluded that the higher the eggshell content, the faster the setting time rate. Using fine CaCO_3_ as a filler can accelerate the cement hydration, especially tri-calcium silicate (C_3_S), which is the major component of cement and provides a nucleation site for the hydration products to precipitate [14,15]. The presence of CaCO_3_ in cement produces calcium carboaluminates when tri-calciumaluminate (C_3_A) reacts with the calcium carbonate (CaCO_3_) [16]. The advantage of using fine filler, like fine CaCO_3_, can lead to low capillary porosity and might also increase early strength [17]. Stark et al. [18] investigated the addition of 6% limestone powder and found that it affected the products of cement hydration and the early strength of cement at 4 days. On the other hand, the addition of limestone powder or calcium carbonate in the cement pastes apparently decreased the setting time [14]. 

There are lots of benefits of using blended cement both in environmental, economical, and technical aspects. The addition of CaCO_3_ to Portland cement is also beneficial as money- and energy-saving concepts. Few studies have investigated the use of eggshell wastes as a partial replacement of cement or as a filler material for cement. This study evaluated the effects of eggshell powder (ESP) on the hydration of cement. Using eggshell powder in engineering aspects will help to reduce solid wastes and environmental diseases. 

## 2. Materials and Methods

### 2.1. Materials

The eggshells were obtained from the Korea Institute of Geosciences and Mineral Resources (KIGAM) institute restaurant in Daejeon, South Korea. The raw eggshells and eggshell powder (ESP) are shown in Figure 2. In the XRD result, the eggshell powder (ESP) shows calcium carbonate (calcite) as its main component, as shown in Figure 3. In this study, type one Portland cement is used. The chemical compositions of the eggshells and type one Portland cement are summarized in Table 1.

### 2.2. Methods

The eggshells were washed properly with tap water and cleansed by deionized water. The sample was then dried at a temperature of 120 °C in an oven for about 2 h. The dried sample was crushed and ground into powder in a steel drum rotating grinder. The crushed sample was sieved through a 75 µm sieve. The dry samples were homogenized by mixing with the help of a spatula for 5 min. In this experiment, hydration studies were carried out on pastes containing 10% ESP (ESP10) and 20% ESP (ESP20) by weight replacement of cement. Ordinary Portland cement (OPC) was also used as a reference. For every sample of pastes, a w/c ratio of 0.4 was used. The samples were put in a separate centrifuge tube, which was conical with dimensions of Ø29 × h115 mm and cured under water at a temperature of 25 °C until the day of testing. The size of each sample was Ø29 × 30 mm. The test was conducted for 1, 14, and 28 days. Samples were put in acetone before testing to terminate the hydration for 24 h and sieved through 75 µm for XRD, thermogravimetric analysis (TGA), and Fourier-transform infrared spectroscopy (FT-IR) analysis.

### 2.3. Characterization

Structural composition analysis was carried out using X-ray diffraction (XRD, Ultima III, Rigaku, Tokyo, Japan) measurement by a graphite monochromator using CuKα radiation (λ = 1.5406 Å) and operating at 40 kV and 30 mA with a scanning speed of 8°/min. The scans were conducted at an angle of 2θ in the range from 5° to 80°. Thermogravimetric analysis (TGA) (Shimadzu DTG–60H, Kyoto, Japan) was also conducted to check the weight losses at different temperatures and the various phase changes in the cement pastes. The samples were conducted at a temperature ranging from 25 °C to 1000 °C at a heating rate of 20 °C/min under nitrogen atmosphere. FT-IR spectroscopy (6700 FTIR, Thermo scientific Nicolet, Waltham, MA, USA) was also conducted for the identification of phases and functional groups. X-ray fluorescence (XRF) (Shimadzu, Kyoto, Japan) was used to find the eggshell composition. The morphology of OPC and the sample containing ESP was analyzed by scanning electron microscopy (SEM) (Tescan Mira 3 LMU FEG, Brno, Czech Republic) with a coater (Quorum, Lewes, UK, Q150TES/10 mA, 120 s Pt coating) and an accelerating voltage of 10 kV.

## 3. Results and Discussion

The investigated compositions of raw eggshells and ordinary Portland cement by X-ray fluorescence (XRF) and the analyzed data are given in the Table 1.

### 3.1. Activity Test

The activity test was performed to check the reactivity of the eggshells when reacted with other materials in the reaction. The test was performed on Graphtec with two channels. Both channels showed the same results. The activity test was performed for 5170 s, and the highest peaks were recorded at 740, 745, 755, 760, 770, and 775 s with a maximum temperature of 52.8 °C. Figure 4 shows the result of the activity test. 

### 3.2. Aging Test

The XRD results of 1, 14, and 28 days are shown in Figure 5, Figure 6 and Figure 7, respectively. In the cement pastes containing ESP, portlandite, calcite, Calcium silicate hydrate (CSH), ettringite, monosulfoaluminate, hemicarboaluminate, and monocarboaluminate were detected. In Figure 5, in the 1-day cement pastes containing ESP (ESP10 and ESP20), ettringite is seen at 2θ = 9.2 and 15.8 with a wide peak due to the addition of ESP. In the 1-day cement paste containing OPC, ettringite is also seen from the silicate hydration reaction. In the early ages of cement containing CaCO_3_, ettringite crystalizes, but in the later ages, the conversion of ettringite to monosulfoaluminate occurs; however, when the amount of CaCO_3_ increases, monosulfoaluminate will change to monocarboaluminate by interchanging sulfate ions using carbonate ions during the hydration of C_3_A [19]. A small peak of hemicarboaluminate and crystallized monocarboaluminate is also found. In the 14-day cement pastes, ettringite is consumed and monosulfoaluminate is found; this is because the stability of ettringite decreases with the addition of ESP. In the 1- and 14-day cement pastes with ESP, small intensities of hemicarboaluminate appear, but in the 28-day paste, it almost disappears and is converted to monocarboaluminate. Hemicarboaluminates are not stable in the presence of excess calcite (CaCO_3_) at a temperature of 25 °C [20]. The existence of monocarboaluminate is visible from the 1 day of hydration with a small peak in ESP10 and high peak in ESP20 and maintains its stability up to 28 days. The existence and formation of the high peak of monocarboaluminate in 1 day might be caused by the presence of more ESP. The higher the percentage of ESP added to the cement paste, the earlier the formation of carboaluminate. Monocarboaluminate is stable because of the slow dissolution of calcium carbonate and its insolubility. Thermodynamic calculation shows that monocarboaluminate, portlandite, and CSH are stable phases in calcite-containing cement [21]. Bonavetti et al. [18] detected the presence of monocarboaluminate in 15 min of hydration and found a high-intensity peak in 24 h with the addition of CaCO_3_. The initial reaction of water with C_3_A can be modified by the use of calcium carbonate where hydrated calcium carboaluminate will rapidly develop as a barrier on the surface of C_3_A grains [22]. The CaCO_3_ particles surround the C_3_A particles and, at the time of contact with water, react to form carboaluminate on the surface of the C_3_A particles [23]. The presence of calcium carbonate in the cement pastes suppresses the conversion of ettringite to sulfoaluminate and facilitates the replacement of sulfoaluminate to carboaluminate [24]. In cement pastes containing CaCO_3_ as a filler, the transition from Aft to AFm phases is modified because of the addition of calcium carbonate to the hydrating system. Portlandite and CSH are formed when tri-calcium silicate (C_3_S) and di-calcium silicate (C_2_S) react with water. Portlandite and CSH are the major hydration products in all cement pastes.

### 3.3. TGA/DTA Analysis

The differential thermal analysis (DTA) and thermogravimetric analysis (TGA) results are given in Figure 8 and Figure 9, respectively. In Figure 8a, TGA results of OPC, ESP10, and ESP20 in the 1-day cement pastes are given. In all cement pastes, different weight losses at different phases are recorded. The first weight loss is recorded in the first phase, which is from 25 to 105 °C. In this phase, the weight loss is due to the dehydration of adsorbed water and the evaporation of moisture as well as water loss from CSH layers and the dehydration of ettringite. A 4% weight loss is recorded in this phase. Further mass loss is also seen after 105 °C, indicating continuous thermal decomposition of complex mixtures [25]. The second phase is where portlandite is decomposed into free lime. A chemical reaction also takes place in this phase. The dehydration of portlandite occurs from 470 °C to 570 °C with a 3% weight loss. The third phase is the decomposition of CaCO_3_, where CO_2_ is released. The temperature where CaCO_3_ decomposes is 700 °C for OPC and ESP10 [26], but for ESP20, it is 740 °C. The temperature difference of ESP20 is due to the higher amount of CaCO_3_ added to the cement pastes than the others. The amount of weight loss of ESP20 in this phase is also higher than the others, which is 9%. In Figure 9a, TGA results of OPC, ESP10, and ESP20 in the 28-day cement pastes are also given. In the 28-day cement pastes, the phases are similar to the 1-day cement pastes, except ESP20. In ESP20, the decomposition of CaCO_3_ starts at 750 °C, and the weight loss increases slightly by 1%. In the DTA curves of the 1-day cement pastes (Figure 8b), all three peaks show that there is an endothermic reaction. The intensities of ettringite and CSH decrease in the 28-day cement pastes of ESP20 compared to the 1-day cement pastes. The wide (U-shaped) peaks for the 28-day cement pastes show the decomposition of the ettringite and CSH phase. On the other hand, the peaks of portlandite increase for the 28-day cement pastes containing ESP10 and ESP20. This result resembles the XRD results. In all cement pastes containing ESP, the degree of hydration is higher than that of OPC in the early stages. For ESP20, the degree of hydration continues up to 28 days, as shown in Figure 10. The addition of ESP in the cement pastes accelerates the hydration of the cement in the early stages.

### 3.4. FT-IR Analysis

FT-IR is conducted to confirm the phase identification of the hydration products of each cement paste. The results of FT-IR for OPC, ESP10, and ESP20 for the 1, 14, and 28 days are given in Figure 11. The band at 3642 cm^−1^ is related to the OH^–^ group associated with Ca(OH)_2_ [27]. The existence of ettringite in the 1-day cement pastes is evident by the vibration of ν3-SO4^2−^ located at 1120 cm^−1^ [28]. In the spectra, monocarboaluminate is also detected by the split ν3-CO_3_^2−^ at 1416 cm^−1^ with ν2-CO_3_^2−^ at 875 cm^−1^ [29]. The intensity of monocarboaluminate is higher for pastes containing ESP20, which resembles the XRD result. The intensity found from 900 to 1100 cm^−1^ is due to the polymerization of silica to form CSH [29].

### 3.5. SEM Analysis

The morphology of ordinary Portland cement (OPC) and the sample containing eggshell powder (ESP) is shown in Figure 11. In both samples, portlandite (plate-like shape) and CSH (sheet-like structure) are the major hydration products. In OPC (Figure 12a), a needle-like structure is visible, which indicates the presence of ettringite. In samples containing ESP (Figure 12b), ettringite is not shown, because of the presence of calcium carbonate in the sample; instead, monocarboaluminate is visible, which is a crystalline structure. Baquerizo et al. [30] also found a crystalline morphology of monocarboaluminate in studies. In the presence of calcium carbonate, the stability of ettringite is very low. The SEM results also resemble the XRD results.

## 4. Conclusions

Eggshell powder is one of the new supplementary cementitious materials (SCMs) and a source of calcium carbonate in the form of pure calcite. Eggshells have been proven to be of good quality to react with other materials by the activity test. The addition of eggshell powder to Portland cement accelerates the hydration reaction by reacting with tri-calcium silicate (C_3_S) and also influences the hydration product of cement paste by providing nucleation sites. The precipitation of CSH is also promoted by the nucleation effect of ESP. Cements with the addition of ESP show a great increase in the degree of hydration starting from the early stages, but OPC shows a low result compared to ESP10 and ESP20. This shows that cement with the addition of ESP has fully reacted with water compared to OPC. The nucleation effect of ESP also improves the degree of hydration. Eggshell powder reacts with tri-calcium aluminate (C_3_A) to form monocarboaluminate. The 20% eggshell powder (ESP20) shows a good result for the formation of monocarboaluminate in all test days of the cement paste and for the acceleration of hydration of the cement pastes. In the FT-IR, monocarboaluminate is also detected by the split ν3-CO_3_^2−^ at 1416 cm^−1^ with ν2-CO_3_^2−^ at 875 cm^−1^. The formation of monocarboaluminate is due to the presence of eggshell powder (ESP). From the results, eggshell powder (ESP) changes the hydration products by involving the hydration process. Monocarboaluminate is stable in ESP20 at a temperature of 25 °C. This might be because of the formation of the strong hydrogen bond in water molecules by the contribution of oxygen atoms from carbonate groups. The slow dissolution of calcium carbonate will also contribute to the stability of monocarboaluminate. Eggshell powder is not only a filler material but also participates in the hydration product of cement. From the point of solid waste recycling, the utilization of eggshells is one of the sustainable, feasible, and cost-effective methods. Eggshells can easily be found in households, chicken factories, and egg breaking factories; thus, from the point of using cheaper materials for cement, eggshells can be an alternative material. It also helps to reduce the emission of CO_2_ from cement factories by reducing the amount of clinker produced and the energy generated in the production of Portland cement.

## Figures and Tables

**Figure 1 materials-12-02483-f001:**
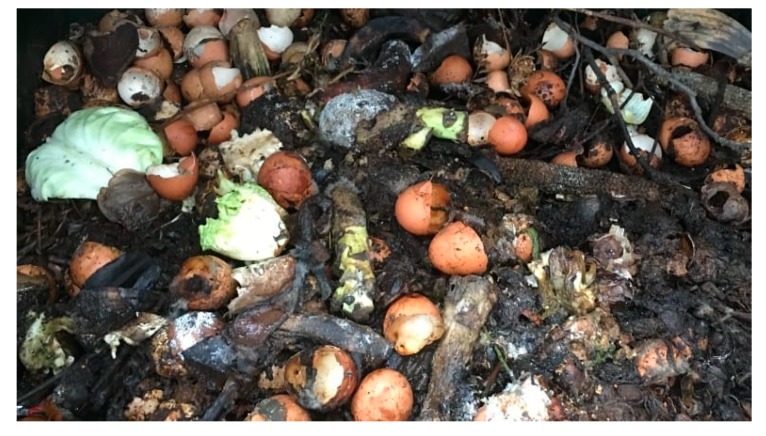
Waste eggshell landfill.

**Figure 2 materials-12-02483-f002:**
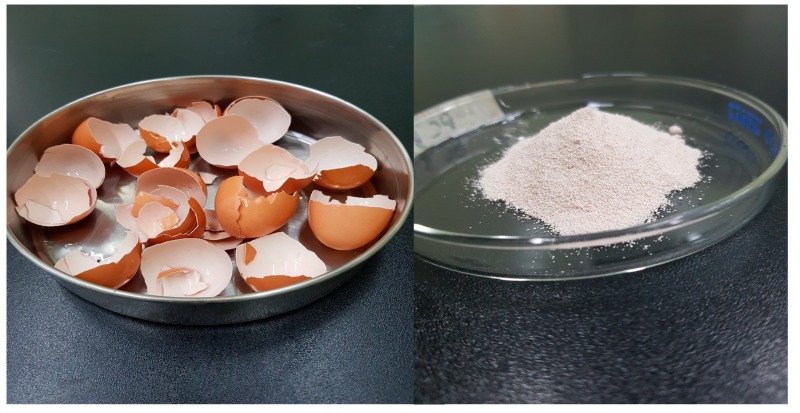
Raw eggshells and eggshell powder.

**Figure 3 materials-12-02483-f003:**
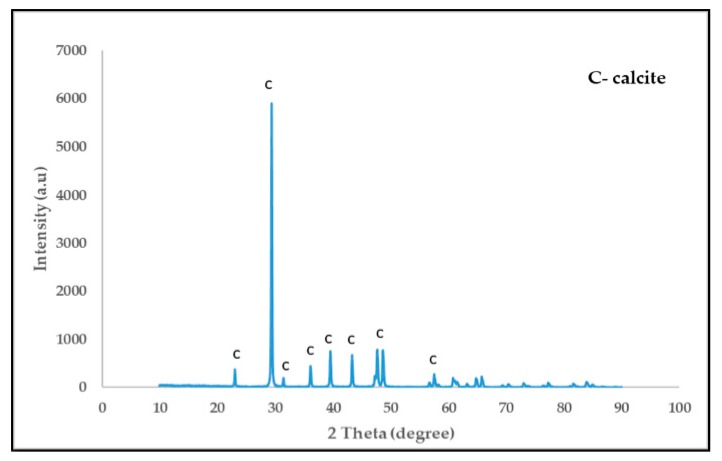
XRD analysis of eggshell powder (ESP).

**Figure 4 materials-12-02483-f004:**
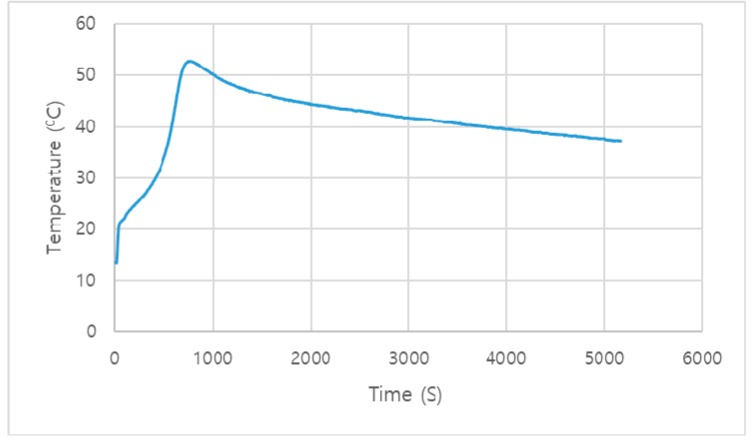
Activity test of ESP.

**Figure 5 materials-12-02483-f005:**
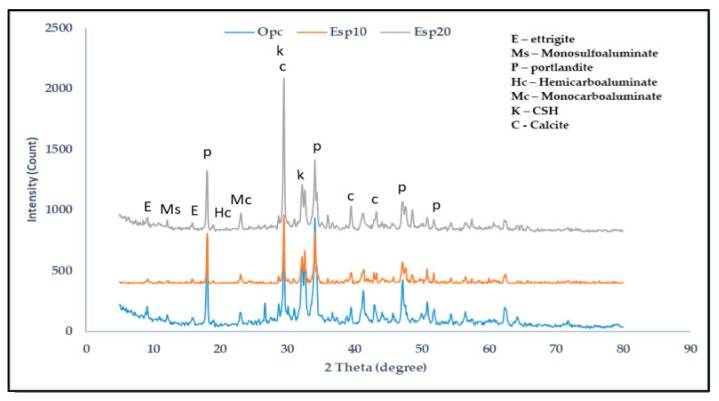
XRD analysis of 1-day cement paste.

**Figure 6 materials-12-02483-f006:**
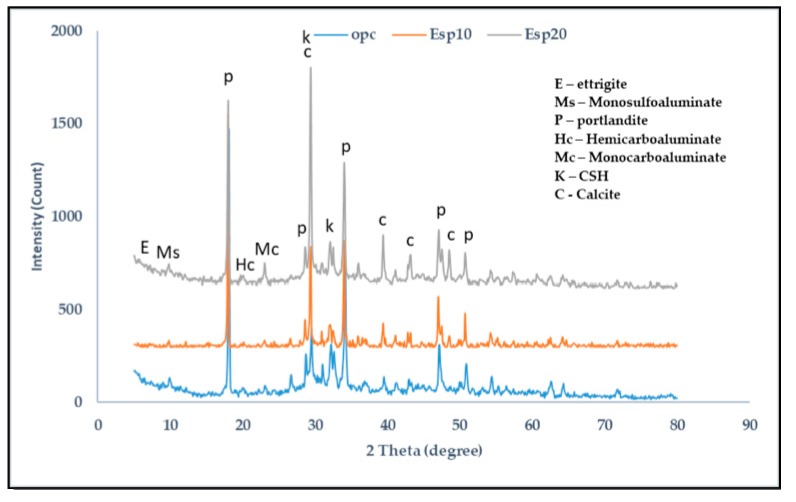
XRD analysis of 14-day cement paste.

**Figure 7 materials-12-02483-f007:**
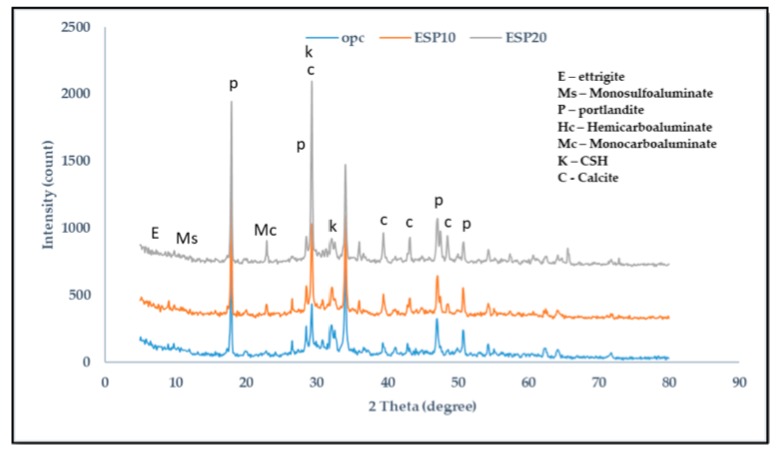
XRD analysis of 28-day cement paste.

**Figure 8 materials-12-02483-f008:**
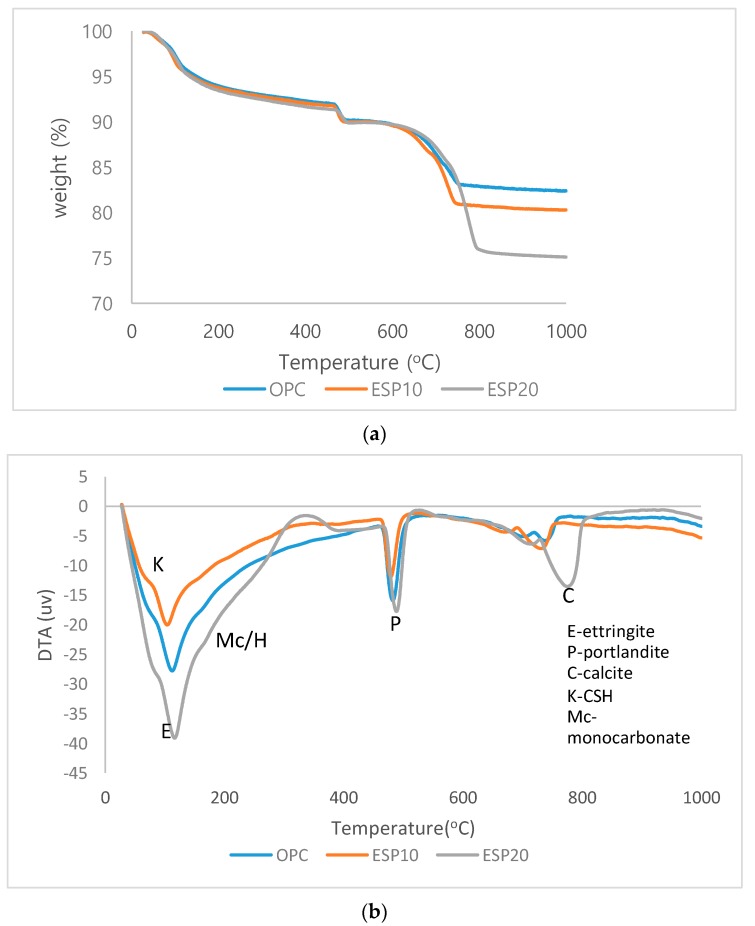
(**a**) TGA result of 1-day cement paste; (**b**) DTA result of 1-day cement paste.

**Figure 9 materials-12-02483-f009:**
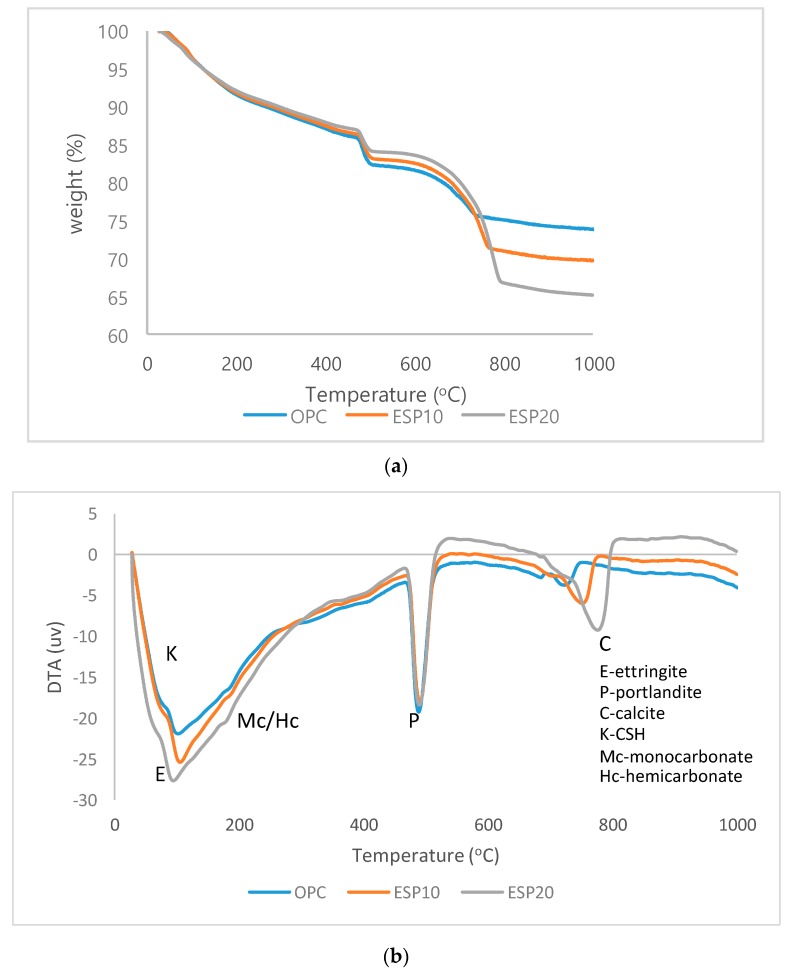
(**a**) TGA result of 28-day cement paste; (**b**) DTA result of 28-day cement paste.

**Figure 10 materials-12-02483-f010:**
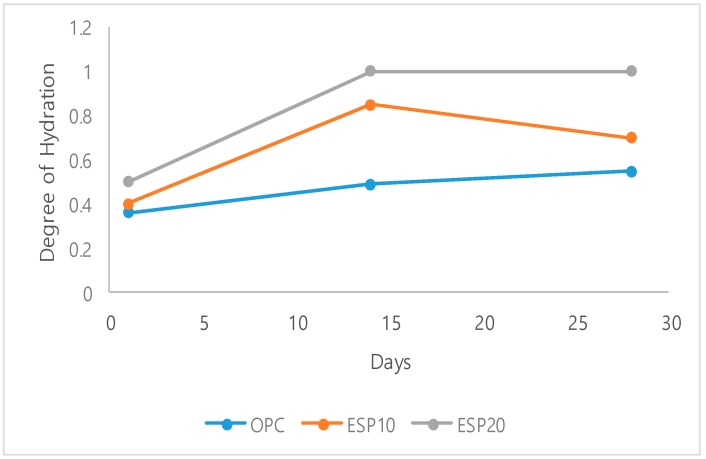
Degree of hydration.

**Figure 11 materials-12-02483-f011:**
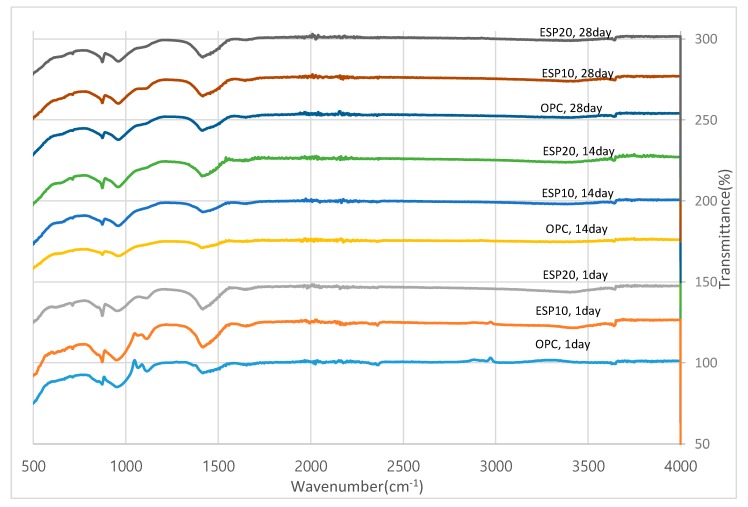
FT-IR results of cement pastes.

**Figure 12 materials-12-02483-f012:**
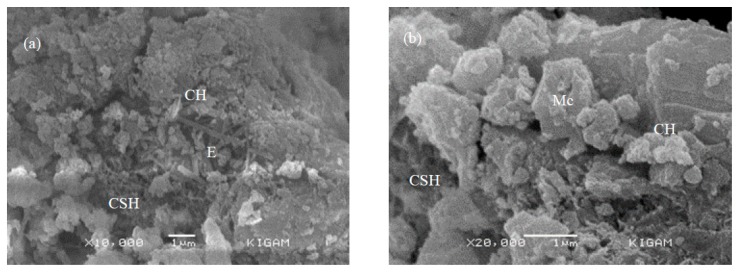
SEM image of (**a**) ordinary Portland cement (OPC) and (**b**) samples containing ESP.

**Table 1 materials-12-02483-t001:** Chemical composition of Portland cement and eggshell.

Chemical Composition	SiO_2_	Al_2_O_3_	Fe_2_O_3_	CaO	MgO	Na_2_O	K_2_O	LOI
OPC	20.99	6.19	3.86	65.96	0.22	0.17	0.60	1.73
Eggshell	0.01	0.01	0.01	52.75	0.51	0.05	0.04	46.62

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
