# Peer review of "Effect of Eggshell Powder on the Hydration of Cement Paste"

_materials, 2019, doi:10.3390/ma12152483_

Round 1
Reviewer 1 Report
The draft presents the properties of a kind of new material, i.e. supplementary cementitious material (SCM), which may be interesting for the reader in civil engineering. Overall the work is well designed, the experimental campaign is consistent and well established. The subject of the paper is an up to date. Additionally, the quality is totally satisfied with requirement of the journal. Therefore this paper could be published. However, there are a few minor issues to deal with:
(1) Eggshell Powder is a new material in the SCMs group. Therefore, it would be necessary to show a picture of how the landfill of this material looks like.
(2) In section 2.1. one should also show a apperance of the Eggshell Powder sample.
(3) The article presents the results of research using the following techniques: XRD, TGA and FT-IR. It would be nice to put SEM pictures showing the structure of these materials. If the authors have such pictures, please include them in the article and comment on them.
(4) Please explain carefully the choice of the amount of Eggshell Powder additive, i.e. 10 and 20%. It seems that it is too small in weight difference. It would be better to use e.g. 10 and 30% or 10, 20 and 30% and then check what the trend in the obtained results is.
(5) Do you plan to study the effect of addition of Eggshell Powder parameters of cement composites after a long period of curing, ie. more than 28 days. Interesting, how changes will occur in the parameters and structure of the paste.
(6) In Section 4, one should give several main conclusions resulting from the content of the article (preferably at several points). Currently, this section is more of a discussion than a conclusion.
(7) In the introduction to the article, the authors describe the ecological aspect of conducted research, e.g. searching for cheap materials that can replace Portland cement; reduction of CO2 emission. This topic has already been the subject of publication in the MDPI journals and other journals. It is therefore required that the authors comment on the results of previous papers. In the Introduction section, the following articles should be discussed and cited:
“The influence of microcrack width on the mechanical parameters in concrete with the addition of fly ash: Consideration of technological and economical benefits. Construction and Building Materials, 2019.
“Effect of fly ash addition on the fracture toughness of plain concrete at third model of fracture, Journal of Civil Engineering and Management, 2017.
Author Response
Answer to the 1st reviewers comments to the article Material-556491
The draft presents the properties of a kind of new material, i.e. supplementary cementitious material (SCM), which may be interesting for the reader in civil engineering. Overall the work is well designed, the experimental campaign is consistent and well established. The subject of the paper is an up to date. Additionally, the quality is totally satisfied with requirement of the journal. Therefore this paper could be published. However, there are a few minor issues to deal with:
Dear Reviewer, thank you so much for your positive and constructive comments.
(1) Eggshell Powder is a new material in the SCMs group. Therefore, it would be necessary to show a picture of how the landfill of this material looks like.
I add a picture of eggshell landfill looks like as your request.(Line 49)
(2) In section 2.1. one should also show a apperance of the Eggshell Powder sample.
I include the raw eggshell and eggshell powder as your request. (Line 87)
(3) The article presents the results of research using the following techniques: XRD, TGA and FT-IR. It would be nice to put SEM pictures showing the structure of these materials. If the authors have such pictures, please include them in the article and comment on them.
I also include SEM image of ordinary Portland cement (OPC) and sample containing ESP and tried to comment on them. (Line 279)
(4) Please explain carefully the choice of the amount of Eggshell Powder additive, i.e. 10 and 20%. It seems that it is too small in weight difference. It would be better to use e.g. 10 and 30% or 10, 20 and 30% and then check what the trend in the obtained results is.
The results that I found with the 10 and 20% ESP gave me enough results about what I want to discuss on this article. Even though the weight difference is small, the results for each percentage is different. so that is the reason why I choose this. But if I was doing mechanical properties of the pastes, the 30% might bring difference on the results.
(5) Do you plan to study the effect of addition of Eggshell Powder parameters of cement composites after a long period of curing, ie. more than 28 days. Interesting, how changes will occur in the parameters and structure of the paste.
No I don’t have any further plan
(6) In Section 4, one should give several main conclusions resulting from the content of the article(preferably at several points). Currently, this section is more of a discussion than a conclusion.
As your request, I tried to discuss more in section 4
(7) In the introduction to the article, the authors describe the ecological aspect of conducted research, e.g. searching for cheap materials that can replace Portland cement; reduction of CO2 emission. This topic has already been the subject of publication in the MDPI journals and other journals. It is therefore required that the authors comment on the results of previous papers. In the Introduction section, the following articles should be discussed and cited:
“The influence of microcrack width on the mechanical parameters in concrete with the addition of fly ash: Consideration of technological and economical benefits. Construction and Building Materials, 2019.
“Effect of fly ash addition on the fracture toughness of plain concrete at third model of fracture, Journal of Civil Engineering and Management, 2017.
I cited and discussed on this two papers as your request. (Line 37-41)

Reviewer 2 Report
This paper reports the effect of eggshell powder on the hydration of cement paste. However, the topic is interesting, but the novelty of work is low and there is not any new finding in this article. For improve the quality of the article, I recommend to do more experiments such as mechanical properties and workability of samples containing ESP. Microstructural studies like SEM or X-ray tomography can help to increase the scientific aspect of this work.
As a minor comment, in section 2.2, what are the dimensions of each sample? and if the samples were put in the plastic bag, how they were in contact with water for water curing?
Author Response
Answer to the 2nd reviewers comments to the article Material-556491
This paper reports the effect of eggshell powder on the hydration of cement paste. However, the topic is interesting, but the novelty of work is low and there is not any new finding in this article. For improve the quality of the article, I recommend to do more experiments such as mechanical properties and workability of samples containing ESP. Microstructural studies like SEM or X-ray tomography can help to increase the scientific aspect of this work.
Dear reviewer, thank you so much for your comments. The reason that I didn’t do mechanical properties and workability test is because, as the title describes I want to focus only on the hydration products formed as a result of the addition of ESP. But as your request, I include SEM results. (Line 269-280)
As a minor comment, in section 2.2, what are the dimensions of each sample? and if the samples were put in the plastic bag, how they were in contact with water for water curing?
I used the word plastic bag as a common name, but specifically I used centrifuge tube, conical-type with a dimension of Ø29 × h115mm, and the dimensions of the samples were Ø29 × h30mm. So since the size of the samples were smaller than the tube, the water can fully contact the sample. (Line 97-99)
Reviewer 3 Report
Dear Authors,
In this study the authors was investigated the effect of eggshell powder on the hydration of cement products.
The work is interesting, but it could have been more consistent
It has specific interesting analyzes and necessary tests but not well explained.
The sections results and discussions and also conclusions could have been more consistent explanations.
Author Response
Answer to the 3rd reviewers comments to the article Material-556491
Dear Authors,
In this study the authors was investigated the effect of eggshell powder on the hydration of cement products.
The work is interesting, but it could have been more consistent
It has specific interesting analyzes and necessary tests but not well explained.
The sections results and discussions and also conclusions could have been more consistent explanations.
Dear Reviewer,
Thank you for your positive and constructive comments. I have tried to discuss more and be consistent in the conclusions part as your requests. (Section 4)
Round 2
Reviewer 1 Report
I have no comments.
Reviewer 2 Report
The authors could address my comments and now the manuscript can proceed to the further steps.